# Microbe-Based Sensor for Long-Term Detection of Urine Glucose

**DOI:** 10.3390/s22145340

**Published:** 2022-07-17

**Authors:** Dunzhu Li, Yunhong Shi, Yifan Sun, Zeena Wang, Daniel K. Kehoe, Luis Romeral, Fei Gao, Luming Yang, David McCurtin, Yurii K. Gun’ko, Michael E. G. Lyons, Liwen Xiao

**Affiliations:** 1Department of Civil, Structural and Environmental Engineering, Trinity College Dublin, D02 PN40 Dublin, Ireland; lidu@tcd.ie (D.L.); yushi@tcd.ie (Y.S.); yisun@tcd.ie (Y.S.); zewang@tcd.ie (Z.W.); gaofe@tcd.ie (F.G.); yanglu@tcd.ie (L.Y.); mccurtid@tcd.ie (D.M.); 2AMBER Research Centre and Centre for Research on Adaptive Nanostructures and Nanodevices (CRANN), Trinity College Dublin, D02 PN40 Dublin, Ireland; kehoeda@tcd.ie (D.K.K.); romerall@tcd.ie (L.R.); melyons@tcd.ie (M.E.G.L.); 3School of Chemistry, Trinity College Dublin, D02 PN40 Dublin, Ireland; igounko@tcd.ie; 4Trinity Centre for Bioengineering, Trinity Biomedical Sciences Institute, Trinity College Dublin, D02 PN40 Dublin, Ireland; 5BEACON, Bioeconomy SFI Research Centre, University College Dublin, D07 R2WY Dublin, Ireland; 6TrinityHaus, Trinity College Dublin, D02 PN40 Dublin, Ireland

**Keywords:** glucose sensor, microbial fuel cell, urine glucose, diabetes pre-screening, selectivity

## Abstract

The development of a reusable and low-cost urine glucose sensor can benefit the screening and control of diabetes mellitus. This study focused on the feasibility of employing microbial fuel cells (MFC) as a selective glucose sensor for continuous monitoring of glucose levels in human urine. Using MFC technology, a novel cylinder sensor (CS) was developed. It had a quick response time (100 s), a large detection range (0.3–5 mM), and excellent accuracy. More importantly, the CS could last for up to 5 months. The selectivity of the CS was validated by both synthetic and actual diabetes-negative urine samples. It was found that the CS’s selectivity could be significantly enhanced by adjusting the concentration of the culture’s organic matter. The CS results were comparable to those of a commercial glucose meter (recovery ranged from 93.6% to 127.9%) when the diabetes-positive urine samples were tested. Due to the multiple advantages of high stability, low cost, and high sensitivity over urine test strips, the CS provides a novel and reliable approach for continuous monitoring of urine glucose, which will benefit diabetes assessment and control.

## 1. Introduction

Diabetes mellitus is a significant public health issue, crippling over 300 million people globally [1]. The urine glucose test is an effective method to screen for prediabetes (a preliminary test before the blood glucose test) and assist in diabetes control [2,3,4,5,6,7]. Currently, enzymatic urine test strips are widely used for urine glucose level testing. This method, however, is subjective to the user’s eyesight, meaning that the results are not highly accurate or reliable. In addition, substances such as antibiotics and ketones and storage conditions (such as temperature change) could interfere with the results, leading to a false-positive or negative outcome [8]. The enzymatic strip is a single-use product, typically made from a plastic ribbon, which results in a significant amount of medical and plastic waste [9,10,11]. More importantly, the enzymatic strip is inconvenient to use on a regular basis. Hence, a reusable, accurate, low-cost, and user-friendly method is needed for urine glucose detection to facilitate diabetes control.

Microbial fuel cell (MFC) technology has attracted a lot of attention recently due to its potential for simultaneously harvesting electric energy and removing pollutants from wastewater [12,13,14,15,16,17]. The current produced by an MFC is determined by the total amount of biodegradable organic matter, and an MFC-based biochemical oxygen demand (BOD) sensor has been successfully developed [18,19,20,21]. MFC sensors have many great advantages, such as high sensitivity, quick response, and low cost. These advantages indicate that the MFC sensor could be a promising glucose detector, especially for long-term continuous diabetes monitoring.

MFCs can generate currents using many different organic molecules. However, MFCs have not been used to detect a specific organic molecule in a complicated liquid mixture, such as human urine. Although typical bacteria are capable of consuming a range of nutrients, they can become extremely picky when more than one carbon/energy source is available. As a simple sugar, glucose is one of the most preferred nutrients for bacteria, which is consistent with the top position of glucose in the nutrient utilization hierarchy [22,23,24]. More importantly, it has been reported that bacteria are likely to block utilization pathways for non-preferred nutrients when glucose is available [22,23]. In addition, a previous study confirmed that the MFC current response profiles of different organic matters were significantly distinct and were dependent on the structure and biodegradability of organic matter [25]. Therefore, it is hypothesized that different glucose concentrations in human urine should lead to different current responses, and MFC could be developed as a glucose sensor.

To validate the hypothesis, a novel MFC-based cylinder sensor (CS) was developed to investigate how the CS’s current response to the glucose level changes. The long-term performance of the sensor was also monitored. As biofilm detachment and accumulation can affect the current generation of MFC-based sensors [19,26,27], the CS in this study was designed to contain a screw bottom to collect and remove the detached biofilm. During the experiment, the CS obtained a quick response, a large detection range, and an excellent linear relationship (R^2^ higher than 0.98) between glucose concentrations and peak currents. More importantly, the CS was able to stably operate for up to 5 months. The selectivity of the CS was assessed by the representative urine organic matter, such as citric acid and urea. Finally, real urine samples from diabetic and non-diabetic individuals were collected and tested to validate the accuracy and reliability of the CS. This study successfully demonstrated the ability of the CS to detect glucose in urine samples. Paired with the low production costs and high stability, the CS could provide an easier and more accessible approach for prediabetes pre-screening.

## 2. Materials and Methods

### 2.1. Sensors’ Construction and Operation

Two identical CSs were developed in the study (Figure 1 and Appendix A). To make the cathode, a piece of stainless-steel mesh was curled to form a cylinder 25 mm in length and 7 mm in diameter (the working area was around 5.5 cm^2^). Then, it was projected by the catalyst mixture (activated carbon, carbon black, and PVDF) described in previous reports [27,28]. One layer of denim fabric was wrapped around the cylinder surface and acted as a separator. Finally, the anode was produced by wrapping 80 mg of graphite fibers around the separator using Ti wire and assembled into the reactor chamber (net volume = 3 mL). This design allowed the detached biofilm to automatically fall into the bottom lid, avoiding the adverse effects of dead biofilm accumulation inside, improving the stability of long-term operation.

Given that flat-cathode MFCs have been widely used in previous BOD sensor studies [18,19], two flat-cathode MFCs sensors (FS) were developed to compare their performance with that of the novel CS (Appendix A). Both FS1 and FS2 were cubic single-chamber reactors with a cylindrical chamber 30 mm in diameter. FS1 was 60 mm in length with a net volume of 30 mL, whereas FS2 was 7 mm in length with a net volume of 4 mL. The anodes were graphite fiber brushes (GFBs) that were 30 mm in diameter and 40 mm (FS 1) and 5 mm (FS 2) in length. All GFBs and graphite fibers were heat-treated at 450 °C for 30 min before being used in this study. All cathodes used in this study were made by the phase inversion method reported previously [28]. Domestic wastewater was used as inoculum. A 5 mM glucose medium was used to culture the biofilm in all sensors unless otherwise noted. The medium was made by adding and mixing the following chemicals with around 0.5 liters of distilled water: KCl, 130 mg; NH_4_Cl, 310 mg; Na_2_HPO_4_·H_2_O, 2.75 g; NaH_2_PO_4_·H_2_O, 4.97 g; vitamin, 12.5 mL; mineral, 12.5 mL [29]. After that, the medium was diluted to 1 liter using distilled water in a volumetric flask. The compositions of the vitamin and mineral stock solutions are listed in Appendix A. The medium was stored in a 4 °C fridge. All sensors were cultured for 1 month before being used in the experiments.

### 2.2. Sensor Characterization

Scanning electron microscope (SEM, Zeiss Ultra Plus) and energy dispersive X-ray (EDX) were performed on cathodes to analyze the topography and structure of sensors. The EDX tests were conducted at an acceleration voltage of 15 kV. Prior to the test, small pieces (~0.25 cm^2^) of the new cathode and used cathode (after 6 months of operation) were cut and cleaned. The biofilm on the used cathode was carefully removed, and the surface was thoroughly washed using DI water. X-ray diffraction (XRD) analysis was also performed using a Bruker D2 Phaser diffractometer over a 2*θ* range of 15 to 85°. Transmission electron microscopy (TEM) analysis was performed on a JOEL 2100 using a 200 KeV acceleration voltage from a Lanthanum Hexborise emission source. The samples were drop-casted onto 300 mesh lacy carbon copper grids (Ted Pella) and allowed to dry overnight.

To characterize the electrochemical performance of the CS, cyclic voltammetry (CV) was conducted using an electrochemical workstation (CH Instruments, Model 680, CHI760D). Referring to previous studies [27,30,31], the anode was connected to the working electrode (WE), an Ag/AgCl (3M KCl) salt electrode was used as a reference electrode (RE), and a Pt counter electrode (CE) was employed for closing the anode. The electrolyte contained 5 mM K_3_[Fe(CN)_6_], 5 mM K_4_[Fe(CN)_6_], and 0.1 M NaCl. To determine the influence of anode biofilm on the performance of the CS, a biofilm-free anode was used as a control. The anode was allowed to operate for 2 weeks after reaching a stable voltage output to ensure complete biofilm formation and acclimatization. During the test, 3 cycles of the CV scan were conducted for each sample, and the 3rd scan was used for the analysis.

### 2.3. Measurement and Analysis

The voltage data was measured using a digital handheld multimeter (USB 6000, National Instruments, Austin, TX, USA), with measurements taken every 0.5 s. Current and other parameters were calculated using basic electrical calculations [32]. All experiments were conducted at a controlled temperature (30 ± 1 °C). All experiments mentioned above were repeated a minimum of 2 times. The urine samples were collected from 4 healthy volunteers and mixed for testing. The glucose concentration of this mixed urine was tested by a standardized analyzer (Konelab 20XT, Thermo Electron Oy, Vantaa, Finland) and was found to be less than 0.2 mM. The diabetes-positive urine samples were prepared by spiking different amounts of glucose in the above-mixed urine and tested by the CS and the Konelab 20XT, respectively.

## 3. Results

### 3.1. Physical and Electrochemical Characterization of Sensor

The surface of the CS’s cathode was observed using SEM–EDX. The cathode surface was rich with pores and fissures, which can benefit oxygen diffusion and promote the current generation of the sensor. The EDX elements mapping and spectra showed that C (carbon) was the most abundant element (accounting for 37.6 ± 3.7% of the total element contents) and was evenly distributed on the surface of the cathode (Figure 2a). This was expected, as the cathode was coated in three carbon-rich materials: AC, CB, and PVDF (formula: -(C_2_H_2_F_2_)_n_-). F (Fluorine) accounted for 10% of all elements in the cathode because it is the second richest element in PVDF. According to the mapping of the elements C, F, and O (oxygen, attributed to AC and CB; Figure 2a), PVDF, AC, and CB were well mixed and formed a uniform surface. TEM imaging also confirmed the homogenous dispersal of the AC catalyst within the diffusion material (Figure 2b). The even mixing of multiple materials within the cathode can reduce the proton/oxygen transfer distance and increase the area for the oxygen reduction reaction (ORR) to occur, promoting the performance of the sensor. The XRD results are shown in Figure 2c. The peak identified at around 20° (2Ѳ) was attributed to PVDF, which is consistent with a previous report [33]. The peaks produced at around 25° and 45° are widely recognized as indicators of activated carbon and carbon black [34,35]. The significant peak at around 45° indicates the high crystallinity of carbon-based material. The high crystallinity of carbon catalysts can promote the electrochemical performance of MFC [35,36].

CV tests were also performed to investigate the electrochemical activity of the CS in electrolytes containing K_3_[Fe(CN)_6_], K_4_[Fe(CN)_6_], and NaCl (for details, see Section 2). The sensor containing the biofilm-free anode showed no significant redox peaks (Figure 2d). By contrast, the sensor containing a biofilm-covered anode exhibited one significant redox couple at around 0.45 V and −0.08 V (vs. Ag/AgCl), corresponding to the anodic and cathodic reactions, respectively. This result indicates that the biofilm substantially improved the electrochemical activity of the CS.

### 3.2. Accuracy and Stability of Glucose Detection with Different Sensor Configurations

Both the FS and the CS had good responses to the change in glucose concentrations (Figure 3). In the range of 0.3–1.5 mM (R^2^ = 0.98, Figure 3a,b), FS1 showed a linear relationship between the peak current and the glucose concentration. When the glucose concentration was greater than 1.5 mM, the peak current became saturated, remaining relatively stable. When the concentration was less than 0.3 mM, no current was detected. When the MFC volumes decreased from 30 to 4 mL (FS2), the detection range expanded to 0.6–2.5 mM (Figure 3c,d, R^2^ = 0.99; there was no detectable current when glucose was 0.3 mM). In addition to the increase in the detection range, FS2 reached the peak current in 10.1 ± 3.7 min. In comparison, FS1, with a large volume, needed around 30 min to reach a much higher peak value when feeding with the same concentration of glucose. The CS, which had the smallest volume (3 mL), had a detection range of 0.3–2 mM (R^2^ = 0.98, Figure 3e,f) and a response time of 6.7 ± 0.4 min, similar to that of FS2.

To investigate the reproducibility, five FS1s were assembled and operated under identical conditions. All five sensors were fed with solutions with glucose concentrations of 2, 0.6, and 3 mM. The relative standard deviations (RSDs) were 3.6% (2 mM), 4.9% (0.6 mM), and 11.7% (0.3 mM), respectively (Appendix A). These RSD values are within the range of the good or acceptable categories reported in previous studies [37,38], which confirms the high reproducibility of MFC regardless of the glucose concentration.

The long-term operation of FS2 and the CS was monitored, as these sensors had shorter response times and larger detection ranges, making them more suitable for sensor development. During the first 2-month operation, the current response of FS2 was stable. However, the current generation and fitting curve’s slope started to decrease in month three (Figure 4a). By the fourth month, the slope of the fitting curve decreased to 10.82 µA/mM, only 19% of the value of month two. By contrast, the current generation and fitting curve of the CS were very stable during the 5-month operation period (Figure 4b). This is likely due to the special structure of the CS. By utilizing the force of gravity, the detached biofilm accumulated at the bottom of the CS lid (9 ± 1.8 mg dry weight/month) and could be washed out after a long-term operation. By contrast, the traditional FS2 design requires a very tiny needle or fine tube to remove the detached biofilm. This method is not efficient and can significantly reduce the net volume of FS2 (from 4 to 3.1 mL after a 3-month operation compared with no volume change in the CS). In addition to volume change, the detached biofilm could also cover parts of the electrodes, resulting in a decrease in current in the third and fourth months for FS2. This relationship between the accumulation of detached biofilm and the current decrease was also reported in a previous study, in which the MFCs had a distance between the electrodes of 5 mm [26]. These results show that the CS is the optimum configuration for long-term glucose monitoring (around 5 months). The CS could not only simplify the maintenance but also reduce the material cost of long-term operation.

Although the voltage generation of the CS remained stable for up to 6 months of operation, the current generation of the sensor decreased from around 110 µA to around 80 µA when fed with 2 mM glucose. SEM–EDX was conducted to check the surface change of the cathode (Figure 4c). In comparison to a new cathode (Figure 2a and Appendix A), a significant blockage or collapsed pores were found on the used cathode surface. The disappearance of pores can limit the oxygen transfer capacity, which reduces the power generation capacity of the sensor. In terms of the surface element composition, the amount of C had significantly increased from 37.6 ± 3.7% to 48.6 ± 1.2% (Figure 2a and Table 1). This is an indicator of cathode fouling, which is consistent with previous studies [39,40]. Evidently, further studies are required to deal with the cathode fouling of the CS.

### 3.3. Selectivity of CS

Urine is composed of many different types of organic matters (such as citric acid, glutamic acid, albumin, urea, and uric acid), which can interfere with the selectivity of the CS to glucose. To characterize the selectivity of the CS, the electrochemical performance differences between glucose and citric acid (the representative interfering organic matter) were investigated (Figure 5). For the CS, the current obtained from 2 mM glucose could reach around 120 µA in 15 min. By contrast, the current generated from 2 mM citric acid only reached 0.4 µA in the same time frame. This was negligible in comparison with the current generated from glucose. When feeding the CS with mixed ratios of glucose and citric acid (typical concentrations in urine, Figure 5b), the current responses were not significantly different from those of the samples containing only glucose (one-way ANOVA test confirmed that the difference was not significant at the 0.05 level).

The current responses of other potential interfering organic components of human urine were also assessed using the CS (Figure 5a). In comparison with the peak current of glucose, there was no detectable current generated by urea and uric acid, while a negligible (less than 5% of glucose in 15 min) current was obtained by glutamic acid, citric acid, and albumin. These results demonstrate that the CS can achieve high selectivity for glucose detection.

### 3.4. Improvement of CS Performance

To improve the performance of the CS, the influence of the culture solution concentration on the current response was investigated. When the glucose concentration in the culture medium decreased from 5 to 1 mM, the current value decreased to an undetectable range (less than 7 µA) within 10 cycles. However, when the glucose concentration increased from 5 to 10 mM, the generated current increased significantly; while maintaining a good linear relationship (R^2^ higher than 0.95 for all) between the current and the glucose concentration (Figure 6a). A noticeable increase in the detection range from 0.3–2.0 mM to 0.3–4 mM was also observed. A further increase in glucose concentration from 10 mM to 20 mM, however, did not improve the performance of the CS significantly. Previous studies have reported that within a defined range, both the biofilm depth and organics degradation ability of the MFCs increase with an increasing organic matter concentration [41,42]. Therefore, using a culture solution containing 10 mM glucose could allow the CS to obtain a higher biofilm activity and better detection performance.

Although the CS has many advantages, such as reusability and stability, the time-consuming capture of the peak current is a significant drawback. In this study, the rate of current increase was analyzed to determine the time it took to reach the peak current. The rate of current increase was obtained by the linear fitting of the current outputs measured in the first 100 s. The CS was fed with the solutions of glucose concentrations between 0.3 and 5 mM, which resulted in very positive correlations between the current increase rates (obtained in the first 100 s) and glucose concentrations (Figure 6b, R^2^ = 0.99). This means that measuring the rate of current increase rather than peak current could result in faster glucose detection (1.67 min vs. 6.7 min).

### 3.5. Practical Application and Cost Analysis

For practical application purposes, the CS’s reliability and accuracy to test real urine samples with unhealthy glucose levels were also assessed. In diabetes monitoring, the patients’ urine glucose levels can vary from 5.5 to 110 mM [43], which is higher than the detection limit of the CS developed in this study (0.3–4 mM). To solve this, real urine samples were diluted around 25 times before testing (Appendix A). To test the accuracy, the urine samples were also tested by a commercial glucose analyzer (Konelab 20XT). All the obtained results are summarized in Table 2. Compared with the glucose analyzer, the percentage recovery of the CS ranged from 93.6% to 127.9% with an average RSD% of 21%. There was no significant difference found at the 95% confidence level using a paired t-test. These results showed good agreement between the CS and the commercial glucose meter. This study demonstrated the reliability of the CS for practical urine glucose monitoring.

The overall material cost of the CS was only USD 0.5 (Appendix A). Due to the long-term stability (5 months) of the device, the test cost of the CS was less than USD 0.01 per use (assuming that each of the CS can be used 150 times test). This is much cheaper than the cost of urine test strips (about USD 0.5 per use). The anode and the chamber accounted for 94% of the cost. As both the anode and the chamber have a long life expectancy (over 2 years) and can be reused, the material cost could be further reduced.

## 4. Discussion

This study demonstrates that an MFC-based sensor can be developed as a low-cost and reliable tool for the continuous monitoring of urine glucose. Evidently, the sensor structure had a substantial influence on the performance of sensors, which is consistent with previous studies on MFCs. The structure can determine the electrodes’ distance, which impacts internal resistance; the electrodes’ surface area, which supports biofilm growth; and the oxygen crossover, which inhibits anode biofilm activity [19,26,32,44]. In addition, we also demonstrated that the accumulation of detached biofilm can significantly reduce the sensor volume. A previous study pointed out that the biofilm accumulation between the cathode and the anode may also result in the short-circuiting of the MFC [26], leading to an error in the test result. Here, we designed a CS structure that allows for effective cleaning and removal of the detached biofilm, resulting in long-term stable performance. It was also noted that the maltose in urine may interfere with the specific detection of glucose given the highly similar structures of the two sugars. Maltose is a disaccharide made of two glucose molecules bound together [45]. The microbe-based sensor may also be capable of degrading maltose and generating a current signal. However, the level of maltose in the urine was relatively low because maltose can be efficiently reabsorbed and metabolized by the human kidney tubular cells of healthy individuals [46]. For instance, a previous study infused a concentration of around 30 mM maltose in adults and found that a concentration of only 0.32 mM maltose was excreted in the urine [47]. Another study also confirmed that no maltose was detected in over 64% of healthy males [46]. However, the contribution from maltose may be high if the individual has other illnesses, such as enzyme deficiencies and inborn errors of metabolism [48]. Evidently, further studies are required to fully understand the potential interference/detection of maltose and the potential of using CS-type sensors to monitor other diseases.

To improve the performance of the CS, further structural modifications, such as increasing the electrode surface area, increasing electrical conductivity, and minimizing the sensor volume can be achieved using novel manufacturing tools, such as 3D printers or light-curing technology. It was noted that the cathode fouling of the CS substantially impacted the CS after the 5-month operation. Hence, further study on the cathode fouling reduction and mitigation is required. There have been multiple effective methods reported in previous MFC studies, such as adding a PVDF bonded separator [49], replacing the cathode binder using polyvinyl alcohol or vanillin [50], and modifying the cathode surface physiochemical properties using Fe-N_4_ [51].

The CS device showed high selectivity to glucose, even when the feeding solution contained a mixture of different organic substrates. This is consistent with the nutrient utilization hierarchy within bacterial communities. Although bacteria are capable of consuming a range of nutrients, they can become extremely picky when more than one carbon/energy source is available. Glucose is well-known as a highly preferred nutrient for bacteria, even within the sugar category [22,23]. For instance, *E. coli* can consume both glucose and lactose. However, when both sugars are available, the bacteria utilize all of the preferred carbon sources (glucose), before using the non-preferred nutrient (lactose) [22,23]. This explains the similar current response when the CS was fed with glucose only and when it was fed a mixture of glucose and citric acid (Figure 5b). Another potential reason for the high glucose selectivity could be a high concentration of glucose-sensitive bacteria growing on the anode biofilm. As the CS was cultured using glucose, there was selectivity for bacterial species that can only utilize glucose for growth and survival. Genetic screening of the anode biofilm to determine glucose-selective bacteria would be beneficial for further modification and improvement of the CS. Although improvement in the sensitivity and accuracy of the CS is potentially achievable when combined with genetic engineering technology, MFC technology exhibits great potential for selectively detecting glucose.

During the use of the CS, there are multiple factors that may influence the current generation and test accuracy. (1) The cathode fouling after long-term operation can decrease the current generation (shown in Figure 4) and result in an inaccurate glucose level (lower than the real value). The anti-bacteria materials, such as nano silver, can be employed to reduce the biofouling at the cathode and extend the stable operation duration of the sensor. (2) Glucose-containing substances, such as maltose, in the urine may generate a current and interfere with the glucose test. Although the presence of maltose may point out other diseases, it is not directly related to diabetes. It is necessary to conduct systematic studies to understand the potential interference. (3) The incorrect use of the sensor also influences the accuracy. For instance, the current generated by the CS can be substantially low if it is placed in an airtight environment with a lack of oxygen. The shortage of oxygen lowers the oxidation speed and the current value generated from the CS. Although more studies are required to fully understand the potential interference, the development of the CS has opened a wide range of opportunities for long-term, continuous urine glucose monitoring. Further development to integrate CSs into the toilet system will allow diabetic individuals to easily and effectively track their condition. Long-term data could be easily uploaded to personal electronics and analyzed through mobile applications. Currently, with improvements in living conditions, the prevalence of diabetes is increasing in many developing countries, such as India and China [52,53]. Early screening is crucial for the management of the disease since many patients acquire diabetes tests only after the symptoms are significant. Implementing low-cost CSs into the toilet system would be an effective method to diagnose and manage diabetes, especially for individuals living in developing countries.

## 5. Conclusions

This study focused on the feasibility of employing MFCs as a selective glucose sensor for monitoring glucose levels within human urine. The novel CS demonstrated a quick response, a large detection range, and an excellent linear relationship between glucose concentrations and peak currents. More importantly, the CS was able to operate stably for 5 months. It was also found that this selectivity could be significantly enhanced to avoid the interference of other organics. In comparison, the CS showed good agreement with a commercial glucose meter (recovery ranged from 93.6% to 127.9%) when diabetes-positive urine samples were tested. Due to its multiple advantages (high stability, low cost, and high sensitivity and selectivity) over urine test strips, the CS provides a novel and reliable approach for the continuous monitoring of urine glucose. This will improve the accuracy and accessibility of diabetes assessment and control.

## Figures and Tables

**Figure 1 sensors-22-05340-f001:**
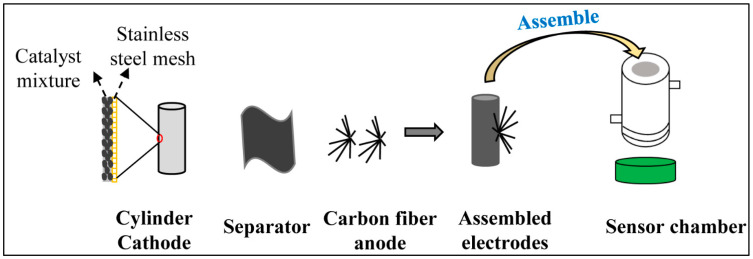
A diagram showing the CS production process.

**Figure 2 sensors-22-05340-f002:**
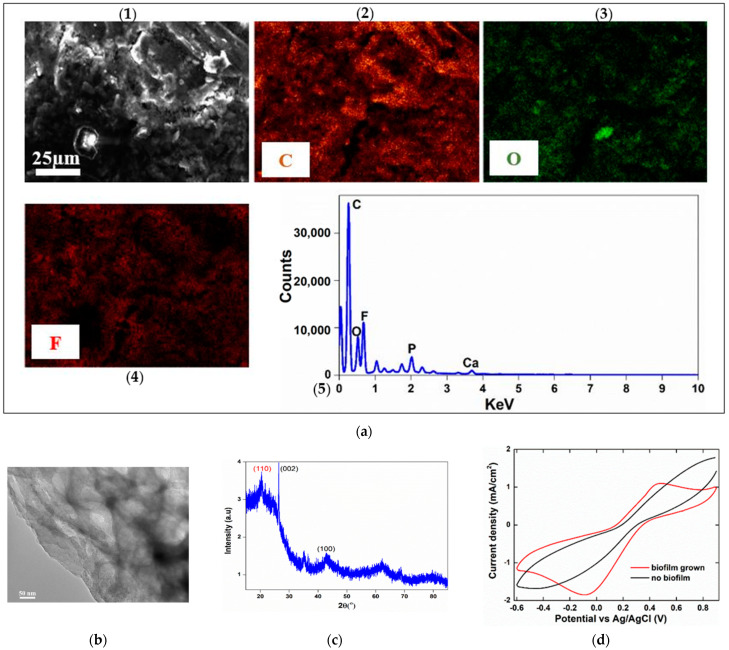
(**a**) SEM images and elements mapping of C, O, and F (**1**–**4**) and full EDX spectra (**5**) of the CS’s cathode. (**b**) TEM image of the CS’s cathode. (**c**) XPS survey spectra—full spectrum of the CS’s cathode. The red-labeling of the peak indicates that it is from PVDF, while black indicates that it is from activated carbon and carbon black. (**d**) CV plots of the CS sensor anode with and without biofilm grown, respectively (scan rate of 10 mV/s).

**Figure 3 sensors-22-05340-f003:**
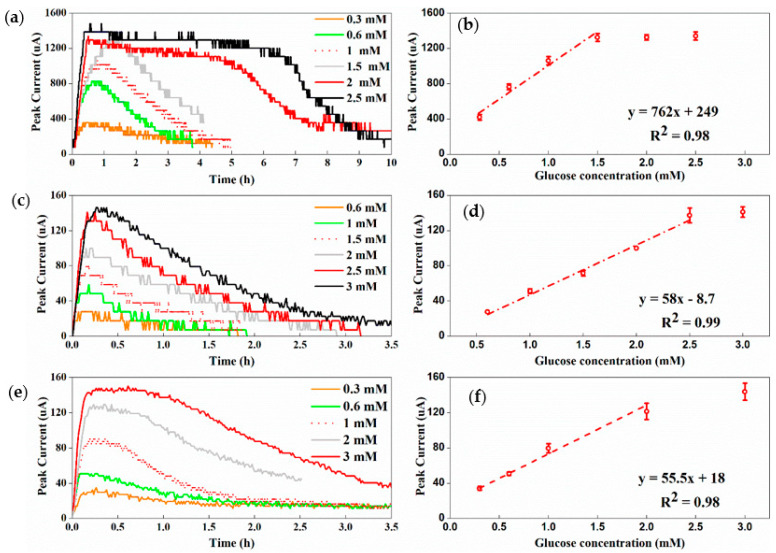
(**a**) Current curves as a response to different concentrations of glucose solutions for FS1; (**b**) the linear correlation graphs between glucose concentration and the peak current of FS1; (**c**) current curves as a response to different concentrations of glucose solutions for FS2; (**d**) the linear correlation graph between glucose concentration and the peak current of FS2; (**e**) the CS current curves as a response to different concentrations of glucose with different concentration; (**f**) the linear correlation between glucose concentration and the peak current of the CS.

**Figure 4 sensors-22-05340-f004:**
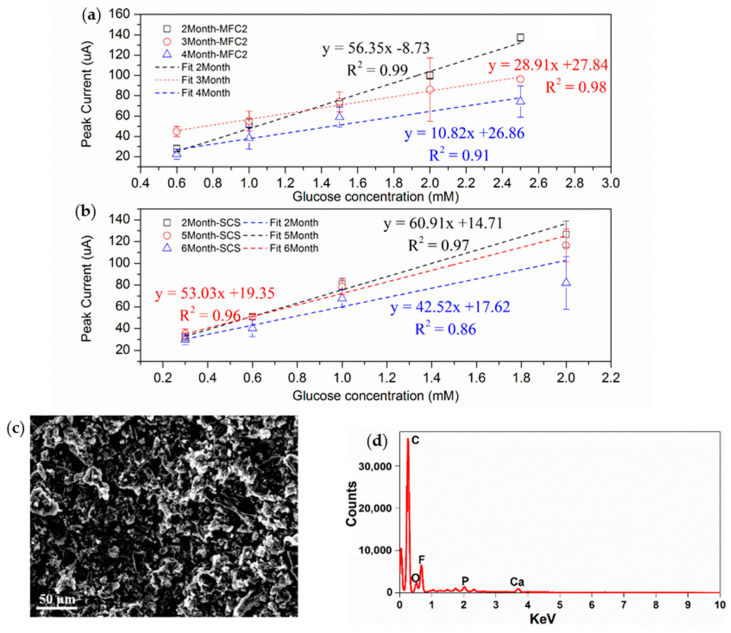
(**a**) Long-term linear correlation between glucose concentration and the peak current of FS2. (**b**) Long-term linear correlation between glucose concentration and the peak current of the CS. (**c**) SEM images of the used CS cathode. (**d**) Full EDX spectra of the used CS cathode.

**Figure 5 sensors-22-05340-f005:**
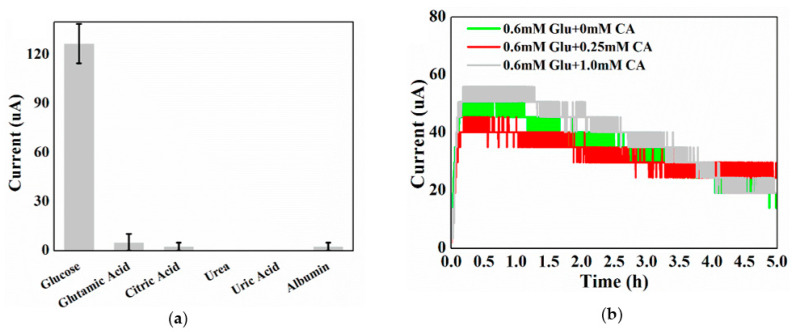
(**a**) Current of the CS in the presence of 2.0 mM representative urine organic matters (albumin concentration = 360 mg/L; obtained in 15 min). (**b**) The current generation of glucose mixed with different concentrations of citric acid (biofilm cultured by glucose; in the figure: Glu—glucose, CA—citric acid). All results were obtained from CS.

**Figure 6 sensors-22-05340-f006:**
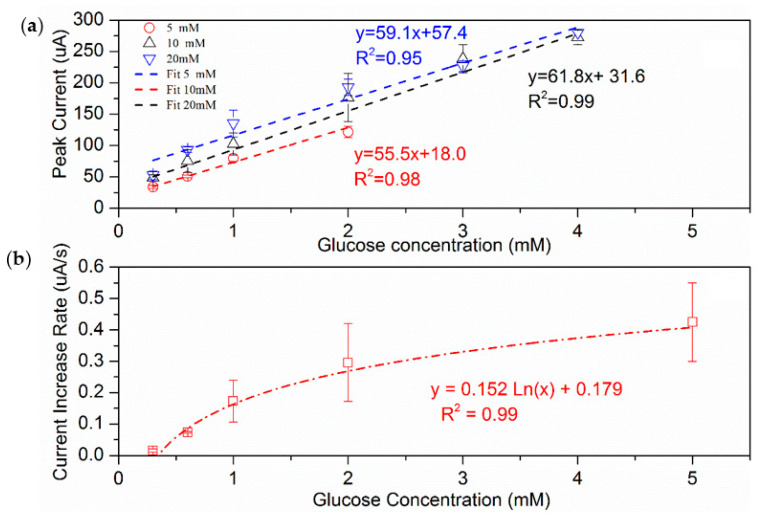
(**a**) The linear correlations (glucose concentration vs. peak current) of the CS with different glucose concentrations; (**b**) the rate of current increase (at 100 s) with different glucose concentrations in the CS (cultured with 5 mM glucose solution).

**Table 1 sensors-22-05340-t001:** Element contents (wt%) of new cathode and used cathode (after 6 months of operation).

Elements	C	O	F	P	Ca
New cathode	37.6 ± 3.7%	8.1 ± 0.2%	9.4 ± 1.4%	5.1 ± 0.2%	1.4 ± 0.2%
Used cathode	48.6 ± 1.2%	4.1 ± 0.3%	9.0 ± 0.04%	3.1 ± 0.1%	1.9 ± 0.3%

**Table 2 sensors-22-05340-t002:** Detection of glucose concentrations in real urine samples using the CS and commercial glucose analyzer.

Samples	Concentration Detected by Analyzer (Glucose, mM)	Concentration Detected by CS (Glucose, mM; N = 3)	Recovery (%)	%RSD
Urine 1	8.4	9.2 ± 0.9	109.4%	16.8%
Urine 2	11.5	10.8 ± 2.8	93.6%	25.7%
Urine 3	18.7	24.0 ± 4.5	127.9%	18.6%
Urine 4	21.5	21.5 ± 4.0	99.9%	18.5%
Urine 5	7.5	8.9 ± 3.9	119.0%	43.4%
Urine 6	22.3	22.2 ± 4.5	99.4%	20.2%
Urine 7	15.3	14.6 ± 3.2	95.7%	21.7%
Urine 8	31.9	34.7 ± 1.9	108.8%	5.5%

## Data Availability

All the raw data are available by contacting L.X.

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
