# Peer review of "Microbe-Based Sensor for Long-Term Detection of Urine Glucose"

_sensors, 2022, doi:10.3390/s22145340_

Round 1
Reviewer 1 Report
Xiao reported work "Microbe-based sensor for long-term detection of urine glucose" is well written can be acceptable in Sensors upon addressing below comments.
1) Some of the recent references related to glucose estimation in real samples were omitted in introduction part.
2) Even though microbial cell is glucose specific, author need to check with maltose which is also another common saccharide found in urine.
3) Interference effects need to be analysed.
4) Authors need to use appropriate notation when they are using equation with concentration. For example here 5 mM K3[Fe(CN)6] + 5mM K4[Fe(CN)6] + 0.1 M NaCl, this looks like garbled.
Reviewer 2 Report
The authors have designed a new sensor for diabetic detection. The language is simple and easy to understand. I am satisfied with the contents, DOE and overall findings. However there are some writing mistakes, typo which I have highlighted in the attached document. please go through to correct them.

Author Response
Point 1: The authors have designed a new sensor for diabetic detection. The language is simple and easy to understand. I am satisfied with the contents, DOE and overall findings. However there are some writing mistakes, typo which I have highlighted in the attached document. please go through to correct them.
Response 1: Many thanks for your review and kind comment. As suggested, we carefully checked the writing mistakes and correct them accordingly.
Please see the revision on Page 10 Line 338, Page 11 Line 365 and Page 12 Line 410.
Reviewer 3 Report
It is a well-design study adding new information to the literature. According to my knowledge, it is a novel paper in its field opening new horizons for further evidence. Authors, succeed to present their findings in a clear way. In addition, the object as well as the results are appropriately discussed in the context of previous literature explaining the importance of the manuscript in its field. Authors succeed to present their data in a clear way adding information to the existing literature.
Therefore, I have no corrections or further work to propose for the improvement of the manuscript and therefore it can be published unaltered.
Author Response
Point 1: It is a well-design study adding new information to the literature. According to my knowledge, it is a novel paper in its field opening new horizons for further evidence. Authors, succeed to present their findings in a clear way. In addition, the object as well as the results are appropriately discussed in the context of previous literature explaining the importance of the manuscript in its field. Authors succeed to present their data in a clear way adding information to the existing literature. Therefore, I have no corrections or further work to propose for the improvement of the manuscript and therefore it can be published unaltered.
Response 1: Many thanks for your review and kind comment.
Reviewer 4 Report
The manuscript by Li et al. describes a microbe-based sensor for the continuous detection of glucose in urine samples. Although the technique described and results produced are very interesting, the report as it currently stands is not of sufficient quality to warrant publication due to the reasons mentioned below:
The manuscript should be revised to avoid few grammatical mistakes present.
In line 108, the authors have to specify the vitamin and mineral being used. They would also need to clarify if the “Vitamin, 12.5 mL; Mineral, 12.5 mL” make up the liter of medium being used or are added to a “liter”. This mainly affects the concentration for repeatability.
Higher resolution figures should be uploaded. The figures are very pixelated.
The statement in lines 196-197, “the detection range expanded to 0.6-2.5 mM” is confusing. It seems that the 0.3 mM condition was simply not tested, and the detection range wasn’t expanded.
In Figure 3, there’s an order of magnitude difference between plot (a) and plots (d)&(f). Why is that? Is this a typo or does it have to do with the volume of sample being used in the analysis?
The units on the y-axis of the graphs have to be consistent. The unit for current is in “mA” in Figure 3 and in “uA” in Figures 4, 5 and 6.
The authors need to clarify their statement in line 219-220 “The low RSD values showed the high reproducibility of MFC regardless of glucose concentration.” The RSD value was almost 12% for one of the concentrations being tested which is not usually considered “low”.
In line 223, the authors make the claim that “During the first 2-month operation, the current response of FS2 was stable.” How often was the device tested? Was it from the same sample and how was the sample stored? What was their acceptable deviation criteria?
In line 247, the authors compare Figure 4c to Figure 2a. It would be easier to create a new figure in the supplementary document and place them side by side and with the same magnification and having the same scale bar to better see the blocked or collapsed pores.
In line 273, the authors claim, “the current responses were not significantly different to the samples” Did they perform any statistical analysis or was is it by observation from figure 5b? The grey points seem 20% greater in magnitude than the red points. Is that an acceptable margin of error during detection?
The y-axis of Figure 5 (b) has 2 axis labels. The concentration label should be removed.
Number of samples being used should be included in table 2 and where applicable in the text.
The paragraph starting at line 357 to line 360 is the instructions and should be deleted.
Round 2
Reviewer 4 Report
The authors have thoroughly improved the manuscript according to initial comments and recommendations provided earlier. The authors are encouraged to address the following very minor recommendations:
Few typos and grammatical mistakes are still present (e.g. “basis vs. base” line 42)
In Figure 3, there’s an order of magnitude difference (1.6 mA vs. 0.16 mA) between plots (a) & (b) and the remaining plots (c) through (f). Authors are encouraged to provide clarification in text on why this difference in magnitude is being observed.
The authors are required to rewrite their statement in lines 106-109 to make it more clear, “The medium was made by adding the following chemicals to distilled water and diluted to 1 litre using distilled water: KCl, 130 mg; NH4Cl, 310 mg; Na2HPO4•H2O, 2.75 g; NaH2PO4•H2O, 4.97 g; Vitamin, 12.5 mL; Mineral, 12.5 mL [29].” What was the original volume of water that these quantities of chemicals were added to before the medium got diluted to 1 litre?
Author Response
Point 1: The authors have thoroughly improved the manuscript according to initial comments and recommendations provided earlier. The authors are encouraged to address the following very minor recommendations:
Few typos and grammatical mistakes are still present (e.g. “basis vs. base” line 42)
Response 1: Many thanks for your review again. As suggested, we asked two English native speakers to proofread the paper. We carefully checked and corrected the writing mistakes accordingly.
Please see the revision on Page 2 Line 42 and Line 65.
Point 2: In Figure 3, there’s an order of magnitude difference (1.6 mA vs. 0.16 mA) between plots (a) & (b) and the remaining plots (c) through (f). Authors are encouraged to provide clarification in text on why this difference in magnitude is being observed.
Response 2: As suggested, we added the content in the main text to clearly state this difference.
Please see the revision on Page 6 Line 209-210.
Point 3: The authors are required to rewrite their statement in lines 106-109 to make it more clear, “The medium was made by adding the following chemicals to distilled water and diluted to 1 litre using distilled water: KCl, 130 mg; NH4Cl, 310 mg; Na2HPO4•H2O, 2.75 g; NaH2PO4•H2O, 4.97 g; Vitamin, 12.5 mL; Mineral, 12.5 mL [29].” What was the original volume of water that these quantities of chemicals were added to before the medium got diluted to 1 litre?
Response 3: As suggested, we revised the content in the main text to clearly state the medium preperation process. It was revised as the following:
The medium was made by adding and mixing the following chemicals to around 0.5 litre distilled water: KCl, 130 mg; NH4Cl, 310 mg; Na2HPO4•H2O, 2.75 g; NaH2PO4•H2O, 4.97 g; Vitamin, 12.5 mL; Mineral, 12.5 mL [29]. After that, the medium was diluted to 1 litre using distilled water in a volumetric flask.
Please see the revision on Page 6 Line 107-110.